# Laser Powder Bed Fusion of Dissimilar Metal Materials: A Review

**DOI:** 10.3390/ma16072757

**Published:** 2023-03-30

**Authors:** Jieren Guan, Qiuping Wang

**Affiliations:** Marine Equipment and Technology Institute, Jiangsu University of Science and Technology, Zhenjiang 212003, China

**Keywords:** laser powder bed fusion, dissimilar metal materials, interface, element diffusion, defects

## Abstract

The laser powder bed fusion (LPBF) technique is used to manufacture complex and customised components by exploiting the unique advantages of two types of metal materials to meet specific performance requirements. A comprehensive overview of LPBF-processed dissimilar metal materials, a combination of different single metals or alloys, is developed. The microstructure in the fusion zone and the corresponding mechanical properties of LPBF-processed dissimilar metal materials are summarised. The influence of processing factors on the mechanism of defect formation, wetting properties and element diffusion behaviour at the interface between different materials and their typical cases are scientifically investigated in detail. Particular attention is paid to energy input, Marangoni convection and interfacial bonding behaviour. The underlying science of the metallurgical structure and properties of the LPBF-processed dissimilar metal materials is revealed. The build quality and efficiency could be further improved by designing machine structures and predicting the process–property relationship. This review provides a significant guide for expanding the industrial application of LPBF-processed dissimilar metal materials.

## 1. Introduction

Based on satisfying the requirements of property-specific areas, multimaterial additive manufacturing (MM-AM) can produce geometrically intricate shapes with desired property variations in a single manufacturing operation [1]. The unique MM-AM method can build designed 3D parts layer-by-layer at exact locations to tailor the composition transition and specific applications. The heterostructure built can broaden the application field of final manufactured components by utilising specific features. Integrating multimaterials provides more options for further design optimisation and performance improvement.

In recent years, multiple materials joining, such as metal–ceramic and metal–metal parts, have been accurately created utilising laser engineering net shaping (LENS) [2,3,4]. The high deposition rate and changed materials in real time make the layered structure of the multiple materials process more efficient and accurate. Consequently, the manufacturing feasibility of functionally gradient materials (FGM) can be realised [2,4]. However, due to the limitations of feed materials and spot sizes, high surface roughness and poor dimensional accuracy have become issues [5]. Although combining subtractive computer numerical control (CNC) machining with additive manufacturing can reduce surface roughness and control geometric accuracy, the tool repair, lack of coolants, and the turnaround time of structures’ predesign limit production efficiency [6].

LPBF technology has attracted abundant investigations and achieved excellent mechanical properties with the increasing demand for metals with high precision and forming efficiency. LPBF is a typical metal-additive manufacturing process that utilises a high-energy laser beam to melt powders track by track and solidify layer by layer to form three-dimensional (3D) components [7]. Little finishing is required to produce shapes and features to achieve a functional form due to the small sizes of the powder particles. Inert gas is often used to prevent contamination and avoid oxidation of the melt pool or hot solidified metal in the forming chamber, such as argon, nitrogen, or helium [8].

In recent years, LPBF-processed single-metal materials have gradually matured. These include aluminium-based alloys [9,10,11,12,13,14,15,16,17,18,19,20,21,22,23,24,25,26], titanium-based alloys [27,28,29,30,31,32,33,34,35,36,37,38,39], copper and copper-based alloys [40,41,42,43,44,45,46,47,48,49,50,51], nickel-based alloys [52,53,54,55,56,57,58,59,60], iron-based alloys [61,62,63,64,65,66,67,68,69,70,71,72,73,74,75,76,77], and various other alloys [78,79,80,81,82,83], and the components fabricated have been successfully commercialised and industrialised. However, due to the machine’s limitation to the use of a single material, research in the field of LPBF-processed dissimilar metals is progressing slowly. By integrating two kinds of dissimilar materials, synergetic enhancement in both strength and fracture toughness could be achieved [84]. The fabrication of dissimilar metal materials satisfies the requirements of harsh circumstances and special functions. The rocket combustor, for example, requires outer materials with high-temperature oxidation resistance and inner materials with excellent heat dissipation. The copper alloy lining and superalloys jacket can be used as a selective combination in the manufacture of rocket burners to take full advantage of their thermal conductivity and mechanical strength. However, the differences in the thermal behaviour of dissimilar metal materials and design factors must be considered to produce a strong and durable bond.

This review provides a comprehensive overview of LPBF-processed dissimilar metal materials and aims to clarify the factors influencing interfacial bonding behaviours. Three types of typical combinations that differ from one metamaterial to another are described in detail. The changes in the microstructure, which are influenced by the process parameters, the inherent characteristics, and the state of the melt pool, are illustrated by introducing different combinations of metal materials. In addition, the corresponding mechanical properties are systematically discussed. Finally, the critical issues and future research directions for LPBF-processed metal materials are provided.

## 2. Classification of Dissimilar Metal Materials

Some possible combinations of dissimilar metal materials can be classified into three categories. Figure 1a presents a sharp transition from material A to B at the interface. In this case, cracking or embrittlement can easily occur at the interface due to physical and metallurgical mismatches and the lack of solubility of the elements in the two materials [85]. Changing the feedstock materials of A and B with a gradient variation may weaken the mismatches and provide a smoother transition, as shown in Figure 1b. This can reduce residual stress concentration and prevent crack propagation. In addition, another material, C, can act as an intermediate layer to prevent the formation of brittle phases and promote wettability between materials A and B, as shown in Figure 1c [86,87]. Studies show that strong metallurgical bonding can be achieved at the interface, though it is still challenging to ensure a defect-free microstructure [88]. Considering the joining fabrication of dissimilar metal materials, preliminary optimisation should be conducted to eliminate or reduce defects, especially near the interface, which drastically affects the bonding strength.

The categories of metal powder materials used in the LPBF process include metals or alloy systems, as shown in Figure 2. When joining different materials, thermal properties, miscibility, and compatibility must be considered at the preliminary design stage. Although they belong to the category of sharp transitions, maraging steel/H13 tool steel bimetals and Al-12Si/Al-3.5Cu-1.5Mg-1Si bimetals have been successfully fabricated with LPBF without cracks, delaminations, or discontinuities [89,90]. When intermetallic compounds (IMCs) or secondary phases occur at the interface of dissimilar materials, cracks form easily since the elastic modulus and coefficient of thermal expansion of the different phases do not match. Brittle IMCs can fracture and cause delamination when joining dissimilar materials [91]. The most common objects are Al/Cu and Ti6Al4V/316L stainless steel (316L or SS) fabrications [92,93]. The interlayer medium is often added to improve bonding and reduce the formation of detrimental brittle phases to avoid the formation of IMCs and minimise their negative influence. The Cu interlayer was introduced to separate the Fe and Ti atoms and bypass the reaction in LPBF-processed Ti6Al4V/316L [93]. Moreover, adding an interlayer can solve the bonding problem between dissimilar materials with a significant melting point difference. Wei et al. used an SS interlayer to successfully bond dissimilar copper/tungsten, taking full advantage of the elements’ wetting behaviour and diffusion mechanism [94]. However, due to the specific limitations of single material usage, most commercially available LPBF machines cannot easily process different metal materials in one step.

In recent years, many researchers have devoted themselves to developing and modifying LPBF machines to fabricate dissimilar material structures. An LPBF system equipped with two upper powder hoppers housing different materials and a lower mixing hopper was operated to fabricate Fe/Al-12Si components. Figure 3a,b show the in-house prototype and the powder feeder. Vibrating plates control the powder feed rate via piezoelectric transducers. Cracking in the microstructure is responsible for the high fragility of the FeAl-IMCs [95]. A Ti6Al4V/IN718 gradient structure was fabricated with experimental optimisation and thermodynamic calculations by applying the feeding hopper and mixing chamber. The brittle Ti_2_Ni phase was responsible for cracking when the IN718 content in the Ti6Al4V powder was more than 20 wt% [96]. Using the ultrasound-assisted multimaterial powder deposition system developed by the University of Manchester, the feasibility of producing LPBF-processed dissimilar materials was demonstrated [97]. Due to the compact structure of a multipowder dispensing feeder, the physical properties can be gradually changed from one material to another to avoid abrupt changes between dissimilar materials [98]. Wei et al. presented a selective multichannel ultrasonic powder delivery system integrated with an LPBF system, as shown in Figure 3c. High-frequency vibrations can dissolve powder agglomeration and ensure the flowability of the powder delivered from the dispensing nozzle. Combined with the vacuum powder removal method, a smooth transition and good metallurgical bonding of 316L/Cu10Sn FGM can be achieved [88,99]. As shown in Figure 3d,f, a series of dissimilar metal structures composed of 316L/Cu10Sn verified the feasibility of LPBF-processed horizontal dissimilar material structures. Furthermore, another method, liquid dispersion powder bed fusion, has been presented to join Inconel 625 and 316L alloys to form hierarchical structures [100]. This approach can potentially develop different metal structures with superior mechanical properties. However, preparing the powder suspension and the predrying process is time consuming and inefficient.

## 3. Microstructure in LPBF-Processed Dissimilar Metal Materials 

When the laser irradiates the surface of the materials, the powders absorb part of the laser’s energy flux, which is converted into heat via electronic interactions with the atoms, increasing the lattice vibrations and consequently raising the temperature of the powders [101]. When the temperature reaches the melting point, the powders melt immediately, forming a molten pool. A high-energy input can provide sufficient thermal activation energy to form unique microstructures and phases [87]. Analogous to the welding process, the molten pool in LPBF appears elliptical and moves along the scan direction. The incident laser scans at a high cooling rate of ~10^7^ K/m [101], which provides sufficient thermal activation energy. The moving heat source moves along the powder bed, and a fusion zone is formed in which the solid metal merges into the liquid metal. Unlike welding, the melt pool is smaller in the LPBF process. The temperature controlled by the laser energy density (*E)* in front of the melt pool is greater than at the rear end. Heterogeneous nucleation occurs at the adjacent boundary of the melt pool, and the base and grains grow towards the interior of the melt pool. There is a certain angle (*ψ*) between the growth direction of the grains and the moving direction of the melt pool, as shown in Figure 4a. The relationship between growth velocity (*R*) and scanning speed (*v*) can be written as *v* = *R*cos *ψ*. When the cos *ψ* is one, *v* is equal to *R*, indicating the fastest growth velocity of the grains. Due to the smaller laser spot and layer thickness, the melt pool and the heat-affected zones are smaller in the LPBF process.

According to classical solidification theory, the solidification mode is mainly determined by the temperature gradient (*G*) and solidification rate (*R*) ratio. As heat is transferred to the deposited substrate, the molten melt solidifies and moves straight towards the molten liquid. Due to the different compositions between the solid and the original liquid, the precipitated solute accumulates in front of the solid–liquid interface. It forms a concentration that leads to constitutional supercooling, as shown in Figure 4b. Both process factors and material properties control grain growth. The criterion for constitutional supercooling can be expressed by:(1)$$\frac{G}{R}<\frac{m_{L}C_{0}\left(1-K_{0}\right)}{D_{L}K_{0}}$$
where *m_L_* is the liquidus slope, *C*_0_ is the initial concentration, *D_L_* is the solute diffusion coefficient and *K*_0_ is the solute equilibrium partition coefficient. With increasing degrees of constitutional supercooling, grain growth successively evolves from planar grains to columnar dendrites and equiaxed grains. The comprehensive influence of *C*_0_, *G,* and *R* on grain morphology is shown in Figure 4c. Columnar and equiaxed grains were frequently observed in LPBF-processed single metals or alloys. The columnar grains are coarse and characterised by anisotropic mechanical properties, while the equiaxed grains are usually small and have more uniform mechanical properties [102].

Due to the influence of rapid heat transfer, a very fine microstructure and a visibly well-connected joint can be observed in LPBF-processed maraging/H13 steel, as shown in Figure 5a,b. The phenomenon of circulating flow was attributed to the irregular distribution of the elements, which is related to Marangoni convection. No secondary phase formed in the transition zone, and various solidification effects contributed to the increase in hardness values at the interface [89]. LPBF can produce a narrow and sound metallurgical bond in an Al–12Si/Al–3.5Cu–1.5Mg–1Si bonding. Four distinct zones developed around the interface and a <001> orientation perpendicular to the top of the Al–12Si part was pronounced, as shown in Figure 5c. The chill effect resulted in texture strengthening, which increased hardness. The results also highlighted the fact that defects, especially pores (Figure 5d), have a stronger influence on failure in the heterogeneous stress distribution in bimetals during LPBF processing [90]. In fact, the high reflectivity of the near-infrared laser beam and the sensitivity to oxygen or hydrogen are also the main reasons for the residual porosity [40,42]. This will be discussed further in Section 5.1.

The dissimilar bonding of steel and copper has attracted many researchers. The interdiffusion of Fe and Cu originating from the welding process can promote metallurgical bonding in the LPBF process. Influenced by the high cooling rates and temperature gradient, the interface showed a strong <111> orientation and an interfacial grain misfit angle of about 21.2° in LPBF-processed maraging steel/Cu, as shown in Figure 6a,b [103]. Chen et al. observed many dendritic cracks near the interface in LPBF-processed 316 stainless steel/CuSn10 structures. As shown in Figure 6c,d, the dendritic crack is perpendicular to the fusion zone and the 316L region, which is attributed to the differences in thermal expansion coefficient and thermal conductivity [104]. A similar conclusion can also be found in the reference [105]. The interfacial transition zones of the 316L/CuSn10 dissimilar materials show microscopic anisotropy, as shown in Figure 7. The average grain size was between the 316L region and the CuSn10 portion. Due to the rapid cooling during the LPBF process, recrystallisation occurred, resulting in local strain and distortion energy. Meanwhile, 316L with 10–13 wt% Ni can be used as an intermediate layer to join W and Cu alloy (CuA). As shown in Figure 6e,f, the mixing of the elements occurs at the CuA–316L interface, which is excited by the circulating flow, and cracks can be observed. However, no pores or cracks were detected at the 316L/W interface, indicating good hydrophilicity [94]. The excellent wettability may explain this between Fe/Ni and W, which will be discussed in more detail in Section 5.2.

In addition, the impact of the IMCs formed must also be considered. Microstructure evolution and joint properties in advanced welding techniques have been extensively studied [106,107,108]. The joint of Al/Cu dissimilar materials illustrates the influence of IMCs. Three typical IMCs, Al_2_Cu (θ), AlCu (η_2_), and Al_4_Cu_9_ (γ_1_) can be formed during welding, and the precipitation process depends on the diffusion reaction of the elements. Due to the lower formation energy, Al_2_Cu (−15.036 kJ/mol) preferentially precipitates and then transforms into Al_4_Cu_9_ (−20.466 kJ/mol) and AlCu (−20.656 kJ/mol) [109]. The reduction in heat input energy modulated by laser power controls the mixing degrees of IMCs [110]. Equiaxed grains consisting of IMCs were observed in in situ Al/Cu alloying by the LPBF process [111]. Due to the high cooling rate of 10^3^ K/s ~ 10^6^ K/s [112], only the Al_2_Cu phase can be detected in the copper alloy AlSi10Mg/C18400 bimetals produced by the LPBF process. Figure 8a shows the proximity of the interface (about 200 μm). Cracks and porosity occur due to the difference in the coefficients of thermal expansion (CTE) and the formation of IMCs. Tensile fracture is a mixture of ductile and brittle modes, with unmelted particles and several voids in the copper region, as shown in Figure 8b [92].

To avoid the formation of IMCs, the addition of an intermediate layer also has a great influence on the microstructure of the interface. Experiments were conducted at different scanning speeds to obtain a stable melt track and a thinner Ti/Cu interface in the LPBF-processed Ti/Cu/Fe structures. Figure 9a shows the Ti/Cu interface microstructure consisting of an ordered L2_1_ phase, an amorphous phase, *β*-Ti + Ti_2_Cu and *α*’-Ti from the Cu to the Ti side. Under external loading, the cracks started in the amorphous phase and preferentially propagated in the *β*-Ti + Ti_2_Cu phase mixture. The tough *α*’-Ti can deflect the crack path towards the Cu interlayer and change the fracture mode from a cleavage surface to a dimpled surface, as shown in Figure 9b. It is worth noting that the micron-sized SS globules and submicro Cu globules distribute within the Cu-rich matrix and SS bands, respectively, due to the liquid miscibility gap in the Fe–Cu system, as shown in Figure 9c,e [93,113]. This phenomenon is also found at the interface of SS–CuA in LPBF-processed W/316L/CuA [94].

## 4. Mechanical Properties of LPBF-Processed Dissimilar Metal Materials

Compared to traditional manufacturing technology, LPBF-processed samples can obtain refined, high-strength solidified microstructures due to ultrafast cooling rates [114,115]. A decrease in grain sizes would lead to an increase in the grain boundary, which is beneficial to nucleation. According to the Hall–Petch relationship, yield strength is inversely proportional to grain size. This can be attributed to the fact that dislocation slip is impeded due to the increasing number of grain boundaries, contributing to the increase in deformation resistance. Affected by the rapid cooling rate in LPBF, a supersaturated solid solution is obtained at the atomic scale [87]. Refinement and solid solution strengthening are the main mechanisms for improving the mechanical properties of SLM-processed single metals or alloys. For example, Si dissolves into an α-Al matrix, forming a network of Al–Si eutectic microstructures and enhancing mechanical properties. As well, Mg_2_Si could precipitate and serve as a reinforced phase [9]. Usually, the precipitation reaction is suppressed due to short interval times. However, artificially extending the interlayer pause time can trigger the precipitation of the nanoscale phases [116].

From the analysis in Section 3, it is clear that for LPBF-processed dissimilar metal materials, excellent interfacial bonding strength and a defect-free interface are critical to achieving ideal mechanical properties. Table 1 lists the mechanical properties obtained for the different types of LPBF-processed dissimilar metal materials. Some researchers found that tensile strength and elongation were intermediate between the two base metals [104,105]. Further optimisation of the process parameters produced a strong bond of 316L/CuSn10 samples with higher tensile stress compared to the CuSn10 samples, as shown in Figure 10a [103]. However, the influence of the building method on tensile strength is also crucial. Vertical and horizontal combinations of 316L/CuSn10 can result in different fracture modes, as shown in Figure 10b,d. The horizontal combination shows a mixed mode of transgranular and intergranular fractures. Some features of herringbone and river patterns can be seen. In contrast, the vertical combination breaks at the junction, which belongs to the cleavage fracture. Some unmelted powders are attached to the fracture surface, indicating that complete melting is a prerequisite. In fracture failure, the crack begins at the stress concentration point. If the interfacial bond is weak, the crack will propagate further. At relatively low stresses, large differences in thermal stresses would increase susceptibility to fracture [96]. Very few, if any, Fe/Cu-IMCs form at the interface. Nevertheless, the solid solution of the elements and the formation of IMCs at the interface can increase hardness.

## 5. Critical Issues for Bonding of Dissimilar Metal Structures

The LPBF technique has allowed the fabrication of intricate features made of metals and alloys with high absorption and low thermal conductivity. Factors contributing to the forming quality for LPBF-processed materials include powder characteristics, alloy types, laser beam focal spot sizes, and other process and design parameters [114]. Spherical particles favour the desired flowability and apparent density. Due to the high susceptibility to oxidation and optically reflective metals and alloys, such as pure copper and aluminium alloys, pores could occur among the interlayers via LPBF. The residual internal stress caused by a higher temperature gradient is prone to result in the formation of cracks and delamination. In this section, some cases of dissimilar metal materials are utilised to explain the critical issues that significantly influence interface bonding. The formation mechanism and control methods of defects, wettability and element diffusion are revealed in detail.

### 5.1. Defects

When the laser beam interacts with the powder bed, complex thermophysical phenomena, including heat conduction, heat convection, and radiation, as well as chemical reactions, occur in the melt pool [120,121], as seen in Figure 11a. Due to the shorter laser–powder interaction time, rapid melting and consolidation make the melt-pool dimensions extremely small [122]. The melting degree and solidification rate of the LPBF process is affected by many factors, such as *E*, scanning strategy, and powder characteristics [123,124,125,126]. The *E* is a combination of laser power (*P*), scanning speed (*v*), hatch space (*h*) and thickness (*t*), which can be written as Equation (2) of [127]. The layer thickness is typically between 20 and 100 μm in LPBF, depending on the particle size distribution of feedstock powders [128,129]. When the applied *E* is insufficient, a lack of fusion can cause porosity. Moreover, shielding gas and air inclusion among powder particles could also result in porosity [130,131,132], as shown in Figure 11b. It is worth noting that nitrogen can dissolve into the liquid melt pool of stainless steel before solidification [133]. If *E* is large during LPBF, high temperature and high recoil pressure in the violent melt pool would cause a splash. The spatters deposited on the solidified surface cause uneven layer thickness, leading to a balling effect [134,135,136]. The balling phenomenon increases the surface roughness and decreases the dimensional accuracy. Worse interlayer wettability leads to poor bonding and incomplete fusion holes [27]. In addition, keyhole formation directly related to metal vaporisation would also generate porosity [137,138,139], as shown in Figure 11c. If *E* is insufficient to completely melt powders, the surrounding particles stick to the surface of the contour tracks, resulting in poor surface quality. Adjustment of energy input could reduce pores and promote bonding between dissimilar materials, which has been verified in LPBF-processed Cu–SS metallic parts [103]. Due to the difference in CTE and thermal conductivity between steel and copper, large residual stresses and misfit strain will inevitably cause cracks at the interface. While increasing *E*, the thermal gradient and residual stresses decrease, contributing to no visible cracks or metallurgical bonding, as shown in Figure 12a,c.
(2)$$E=\frac{P}{v\times h\times t}$$

From the perspective of incident energy, capturing the transient temperature change through the experimental process is difficult. Currently, numerical simulation is introduced to analyse temperature evolution behaviour in several tens of micrometres to hundreds of micrometres [134]. To understand the defect formation at the interface of LPBF-processed Invar36/Cu10Sn dissimilar materials, computational fluid-dynamic and discrete-element method simulation is conducted to explain the hybrid melting–sintering phenomenon with different composition ratios and *E*. The unmelted particles and cracks were present due to the significant differences in the thermal properties. Increasing the Cu10Sn fraction would decrease the depth of the melt pool and insufficient melting of the Invar36 alloy [140].

During the layer-by-layer building process, the direction of the heat flow transfer is from top to bottom. The temperature of the top layer is highest, and the substrate decreases to preheat or room temperature. The temperature gradient leads to residual stress in the LPBF-processed parts after rapid solidification. The residual stress leads to the formation of microcracks and distortion [141]. In contrast, residual stresses and the balling effect can be controlled by adjusting the scanning strategy [142]. Different scanning strategies affect not only the thermal gradient and the direction of heat flow but also the microstructure of the solidification and the mechanical properties [143,144]. The selection of scanning strategy determines the shape and size of the melt pool and the grain structure’s evolution. Random orientation of the scanning direction weakens the texture intensity along the build-up direction and is deemed to reduce residual stress [145]. The interlayer staggered, and island scanning strategies were used to form a 316L/CuSn10 bimetallic structure [104]. The island scanning strategy can reduce residual stress with a shorter scanning vector [146]. Zhang et al. introduced remelting scanning at the interface of the CuSn/18Ni300 bimetallic structure, and lower elongation and tensile stress can be achieved [117]. However, remelting scanning is usually adopted to eliminate defects and improve surface roughness [147,148,149]. At the same time, remelting scanning is beneficial for obtaining a fine microstructure and ideal mechanical properties [150,151]. Therefore, optimising the scanning strategy and process parameters for LPBF-processed dissimilar metal structures is necessary.

### 5.2. Wettability

The wettability discussion is often cited to reveal the balling effect and interlayer adhesion in LPBF-processed metals or alloys [61,152,153]. The balling particles would hinder the movement of the scraper and worsen the wetting characteristics, especially when the atmosphere’s oxygen content is higher than 0.1% [154]. The contact angle can be introduced and expressed by Young’s relation to describe the degree of wetting [155]:(3)$$\sigma_{SG}=\sigma_{SL}+\sigma_{LG}\cdot \cos\theta$$
where *σ_SG_* and *σ_LG_* represent the surface tension between the solid–gas interface and liquid–gas interface, respectively. Expression (3) is established on a flat and smooth surface, as shown in Figure 13a. The wettability between the next layer and solidified layer depends on the surface roughness of the melting and spreading behaviour. However, the melt pool surface has the characteristics of certain fluctuations formed after solidification, which is not completely flat and smooth. The increase in the surface roughness coefficient leads to an increase in surface free energy [156], resulting in poor wettability (Figure 13b). Appropriate control of the energy input and surface treatment can improve the surface quality and increase the adhesion between layers [52,145,157]. The remelting treatment can remelt the small metal spheres, which turn into a melt pool and favourably wet the surface so that the balling phenomenon does not occur [154].

The melt pool should penetrate sufficiently to the previously deposited layer to ensure proper bonding and avoid fusion voids. The depth of penetration of the molten metal into the previous layer is determined by favourable wettability and heat transfer [158]. Due to the large melting point difference and the joining of dissimilar metal materials (e.g., W and Cu), the energy used to melt Cu cannot completely melt the preprinted W. Cu tends to agglomerate into spheres due to surface tension and cannot spread on the W surface, as shown in Figure 13c,d [94].

The energy input affects the temperature gradients between the centre and the edge of the melt pool, as shown in Figure 14a. As the energy input increases, the viscosity of the melt decreases, which promotes flowability. The relationship between viscosity (*μ*) and temperature (*T*) can be defined as follows [159]:(4)$$\mu =\frac{16\gamma }{15}\sqrt{\frac{m}{kT}}$$
where *γ* is the surface tension, *m* is the atomic mass and *k* is the Boltzmann constant. A lower viscosity leads to higher instability of the capillary [160]. A steep temperature gradient would cause surface tension gradients and the resulting Marangoni convention. Intense convection promotes liquid diffusion and improves the metallurgical bonding ability between dissimilar materials. As shown in Figure 14b, Marangoni convection at the interface was enhanced by the high thermal conductivity of copper in the maraging steel–Cu dissimilar metal structure and contributed to the diffusion of the elements. As a result, the bonding strength can be achieved [103]. It can be concluded that the circular flow generated by Marangoni convection can improve element distribution and modify performance.

### 5.3. Element Diffusion

As can be concluded from the results of LPBF-processed metal structures, element diffusion can contribute to interfacial bonding and element mixing in the melt pool [94,103]. The rearrangement of elements induced by Marangoni convection can be verified in the simulation of LPBF-processed Cu10Sn/Inconel718 [161]. A narrow defect-free element diffusion interface can be observed in LPBF-processed Ti5Al2.5Sn/Ti6Al4V with metallurgical bonding [119]. According to Fick’s law, the element diffusion coefficient (*D*) correlates with temperature (*T*) and activation energy (*Q*), which is defined as follows [162]:(5)$$D=D_{0}exp\left(-\frac{Q}{RT}\right)$$
where *D*_0_ is constant, and *R* is the molar gas constant (8.314 J/mol-K). As *T* increases or *Q* decreases, the diffusivity of the elements is enhanced. As diffusion progresses, solid solutions or IMCs form due to the reaction of the elements with each other [92,93,163,164].

To check whether temperature affects diffusion, Wit and Amsterdam used an alternating material-deposition method based on the LPBF process to bond IN718/ SS dissimilar structure. The homogenisation heat treatment visibly promotes element diffusion, resulting in a smooth transition at the interface. For a gradual material transition, the elongation of the diffusion zone is favourable for reducing mechanical-stress concentrations [165]. Since the heat transfer in the melt pool is extremely fast, the element diffusion time is shorter than in conventional fusing processes or joining methods [164,166,167]. The thickness of the diffusion layer at the interface becomes thin, and the precipitation reaction is usually suppressed [168,169]. Guan et al. investigated the mechanism of element diffusion at the LPBF-processed Al/Cu dissimilar metal-structure interface. Only the Al_2_Cu phase was detected at the narrow interface. The vacancy mechanism dominated by the Kirkendall effect affects element diffusion, as shown in Figure 15 [170]. In the LPBF process, the large energy input can drive element diffusion. Nguyen et al. reported that the thickness of the intermetallic layer increased with *E* [171]. The extension of the melt pool lifetime leads to the thickening of the reaction layer and an increase in the IMC content. Therefore, the interfacial bonding of dissimilar metal structures should fully consider the process controlled by the diffusion behaviour of the elements.

## 6. Future Trends and Perspectives

### 6.1. Machine Research and Development

In the LPBF process, the deposition of dissimilar powders can be divided into different powdered layers and the same powder layers [172]. A machine based on spreading powders can only manufacture different layers of metal materials. The premise is to develop more than two types of feeding structures. However, cross-contamination of dissimilar powders cannot be avoided, which increases the cost of using powders. Depending on the different physical properties of the metal materials, the different particle size distributions, spherical levels, and surface roughness can be used to separate dissimilar metal materials [173,174]. However, certain particle-size ranges and the sound-flowability of the powders are required to ensure built quality.

Adjusting the powder feed structure is also crucial for the same powder layer. A horizontally-graded powder layer was spread via a modified powder-mixing hopper, as shown in Figure 16. Uniform and continuous variations in chemical composition and mechanical properties can be obtained for the CoCrMo/In718 samples. This unique method could be adopted to manufacture dissimilar metal materials [175]. Another method employs ultrasonic and vibration-assisted powder delivery systems [88,99]. The hybrid method combining recoated spreading, vacuum suction, and ultrasonic deposition can produce dissimilar metal materials with spatial spacing. However, deposition efficiency and build quality are limited by the size of the parts and remain a challenge.

Large parts are difficult to produce with commercially available LPBF equipment. Therefore, new LPBF machines equipped with specific software should be designed to tailor the requirements of industrial applications in the production of dissimilar metal materials. By designing and optimising the machine, two types of materials were deposited alternately—track by track and layer by layer—to form layerwise structures that could achieve strength-ductility synergy. Spatially heterostructured materials of AISI 420 SS and maraging steel powders transported through two feeders were deposited. A much higher strength was achieved than lamellar materials and linear FGM. The integrated property could surpass the constitutive materials due to the reinforcement mechanism of the rule of the mixture and heterodeformation-induced strengthening [86]. This is a novel approach to solving the problem of the trade off between strength and ductility in LPBF-processed components.

### 6.2. Fabrication of Dissimilar Metal Materials

In a broad sense, *E* directly impacts the powder’s fusion and thus determines the final solidification microstructures. The cyclic thermal history between the overlap of the molten tracks and the interlayer is extremely complicated and cannot be captured in real time. Simulation of the LPBF process using finite element models can capture temperature variations at the microscopic scale but is based on constant thermophysical properties and process parameters. When simulating dissimilar metallic materials, it is difficult to accurately predict the thermal gradient behaviour due to the changes in physical properties induced by the high-temperature range, such as specific heat, thermal conductivity, viscosity, laser absorptivity, etc. In fact, *E* is influenced by *P*, *v*, *h,* and *t* and the scanning strategy used. Due to the rotation angle between the layers and the different scanning methods, the grains’ morphology and growth directions vary. The complex thermal cycle between the solidified and penetration layers leads to a renewed heat treatment, which has desirable and undesirable effects on the microstructure [176]. Microstructural evolution is crucial for the final mechanical properties of dissimilar metal materials. However, experimental data are insufficient to determine general trends for strength as a function of *E*.

With the experience and knowledge gained, the machine learning method was used to overcome the shortcomings of the conventional trial-and-error testing methods and to optimise processing parameters to increase research time and economic costs. Machine learning is an approach that can find suitable LPBF parameters and avoid experimental costs [177]. Machine learning does not require human interaction to label data, achieving a substantial influence from the design and fabrication process to qualification. The process–property relationship can be provided in a zone with composition gradients of 316L Cu parts using machine learning based on a design and surface-roughness-prediction algorithm for the given data [178]. This technology, complemented by high-throughput experiments, is promising for a better understanding of the LPBF process in dissimilar metal materials.

In addition, combining beam shaping and lasers in different wavelength ranges is beneficial for fabricating dissimilar metal materials. Since the laser applied in the LPBF process is usually a Gaussian-shaped beam, high peak intensities would cause defects such as keyholing and spattering, negatively affecting the mechanical properties [139,179]. A pair of identical axicons were used to modify the initial Gaussian beam to produce Bessel beams, as shown in Figure 17. The dispersed energy stabilises turbulence in the molten pool and reduces the defects, contributing to the high density, reduced surface roughness, and robust tensile properties of LPBF-processed 316 SS [180]. If the wavelength range can be matched to the laser absorption of different metallic materials, it is expected that dissimilar components with defect-free and good interfacial joints and excellent service properties can be produced based on Bessel beams.

### 6.3. Application Fields

The future potential applications of LPBF-processed dissimilar metal materials are partially summarised in Table 2. The performance of the fabricated components would be further enhanced by combining the advantages of different physical characteristics. In addition, biomedical and marine shipping should be explored using various dissimilar materials to broaden the application areas. This requires dissimilar materials, such as Ti/Ta and Ti/Al, to have excellent biocompatibility and corrosion resistance.

## 7. Conclusions

LPBF technology promotes transforming and upgrading conventional techniques to smart manufacturing and promises great freedom in fabricating single metal materials with a high precision. Mature manufacturing achievements of LPBF-processed single metals or alloys enable heterogeneous bonding of dissimilar material structures with site-specific functional requirements. Careful consideration of the influence of the thermophysical properties of the materials, process parameters, melt pool formation, and microstructural evolution on the ultimate joint strengths and service performances of LPBF-processed dissimilar metal materials are summarised.

(1) Three types of interfacial microstructures and relevant mechanical properties of dissimilar metal materials were found, including a sharp transition, a gradient transition, and an introduced intermediate layer. The interfacial bonding behaviour is strongly influenced by thermophysical properties and defects, such as porosity, temperature gradients dominated by residual stresses, and mismatch cracks caused by the formation of IMCs;

(2) The combination of different metals or alloys could be integrated using trial-and-error and numerical simulation process parameters. The fundamental processing factors and scanning strategies interact to determine the solidification microstructure and residual stress. The reduction and elimination of defects can be achieved by an appropriate energy input. The change in laser energy density (*E*) has a major impact on Marangoni convection and the melt pool dimensions. Excellent interfacial bonding depends on good wettability between dissimilar metal materials and the diffusion behaviour of the elements. A higher *E* promotes the flow intensity of Marangoni convection and thus contributes to the interdiffusion of the elements. However, for two materials that tend to form IMCs, the energy input should be adjusted based on defect control;

(3) The structural design and system optimisation of spreading, transportation, and separation of the various powders of LPBF machines can accelerate industrial production. The deposition efficiency and the type of intermittent deposition fashion require more attention. The introduction of machine learning and laser-beam shaping can predict the process–property relationship and improve the build quality of LPBF-processed dissimilar metal materials. By fully exploiting the characteristics and advantages of different materials, the performance requirements of complex components for challenging or functional conditions, from aerospace to automotive, could be improved.

## Figures and Tables

**Figure 1 materials-16-02757-f001:**
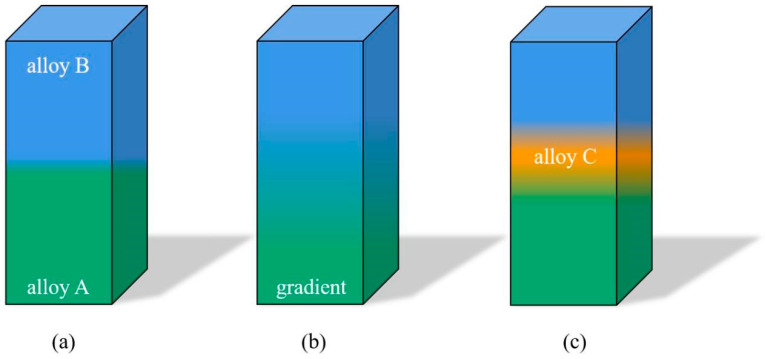
Schematic of dissimilar metal materials structure—different colours represent various metals or alloys, (**a**) represents a sharp transition; (**b**) represnets a gradient variation; (**c**) represents an intermediate layer added.

**Figure 2 materials-16-02757-f002:**
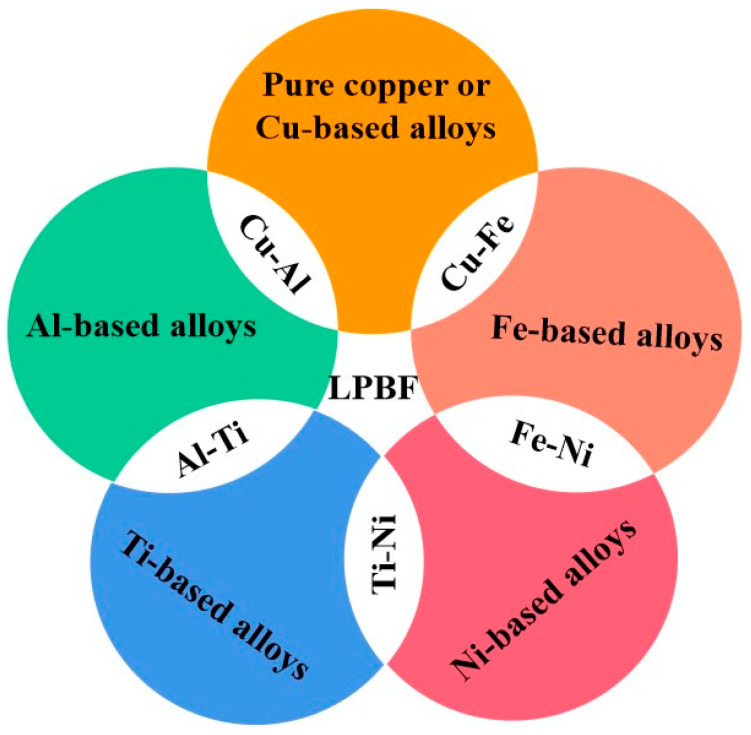
The classification schematic of LPBF-processed dissimilar metal materials.

**Figure 3 materials-16-02757-f003:**
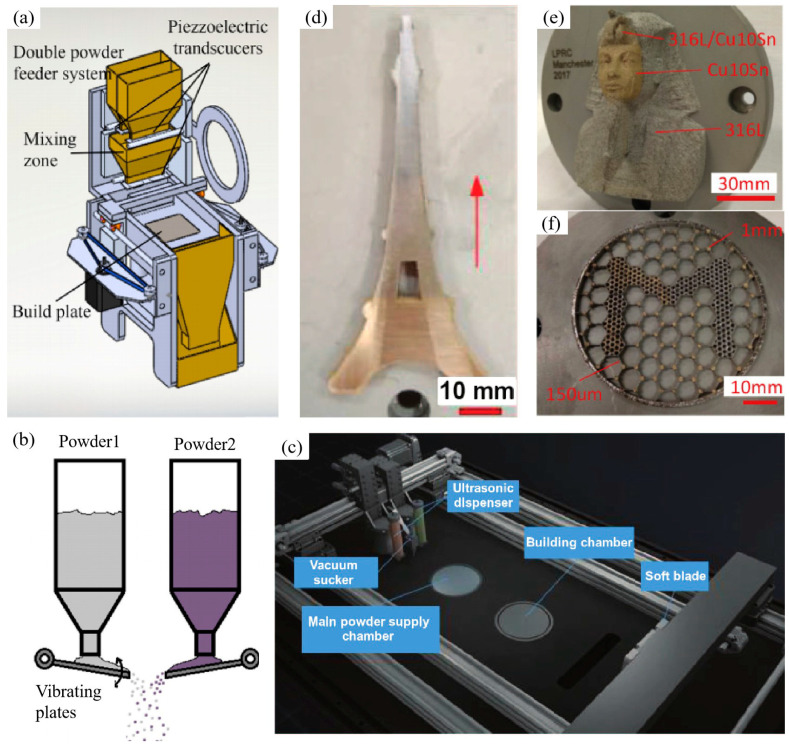
Powder feeder system: (**a**) Prototype schematic diagram; (**b**) Working principle for mixing powders [95]; (**c**) LPBF experimental setup with an ultrasonic powder delivery system; (**d**–**f**) Cu10Sn/316L FGM components [88,97].

**Figure 4 materials-16-02757-f004:**
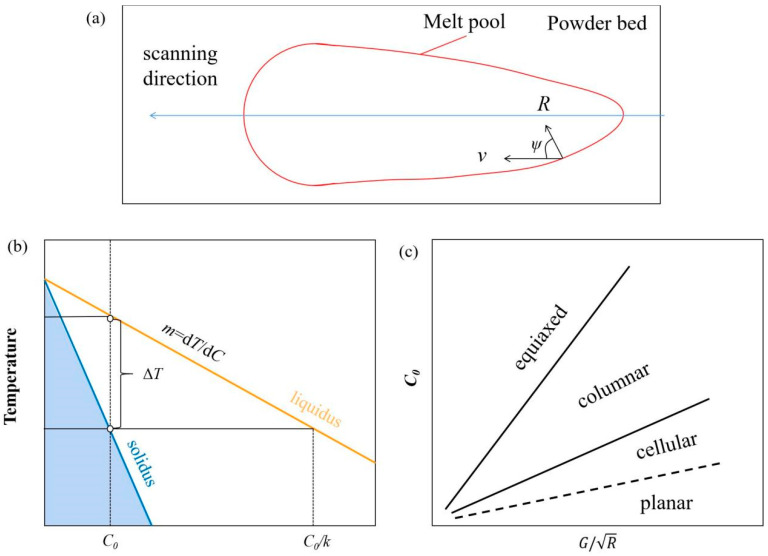
(**a**) Diagram of the relationship between *R* and *v*; (**b**) Schematic representation of constitutional supercooling based on a typical phase diagram; (**c**) Influence of interaction with *G*, *R*, and *C*_0_ on grain morphology.

**Figure 5 materials-16-02757-f005:**
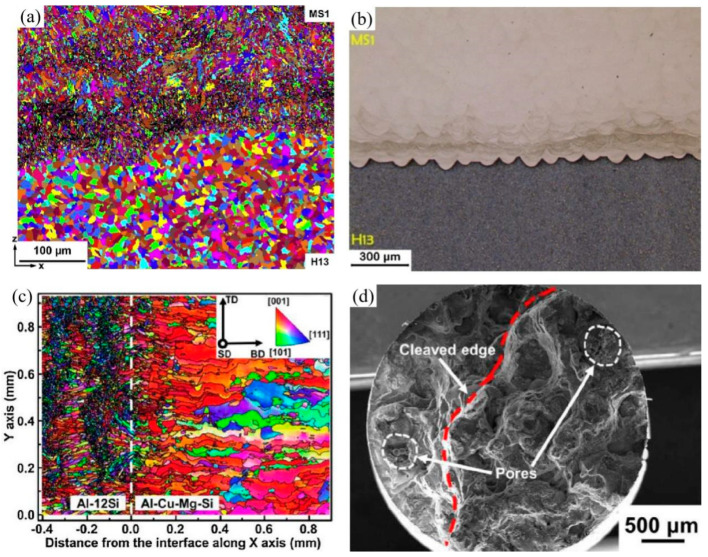
(**a**) Electron backscattered diffraction (EBSD) inverse pole figure (IPF) map; (**b**) Optical micrograph across the interface of the maraging steel–H13 sample [89]; (**c**) IPF of the Al–12Si/Al–3.5Cu interface along the build direction; (**d**) Fracture morphology of the tensile sample [90].

**Figure 6 materials-16-02757-f006:**
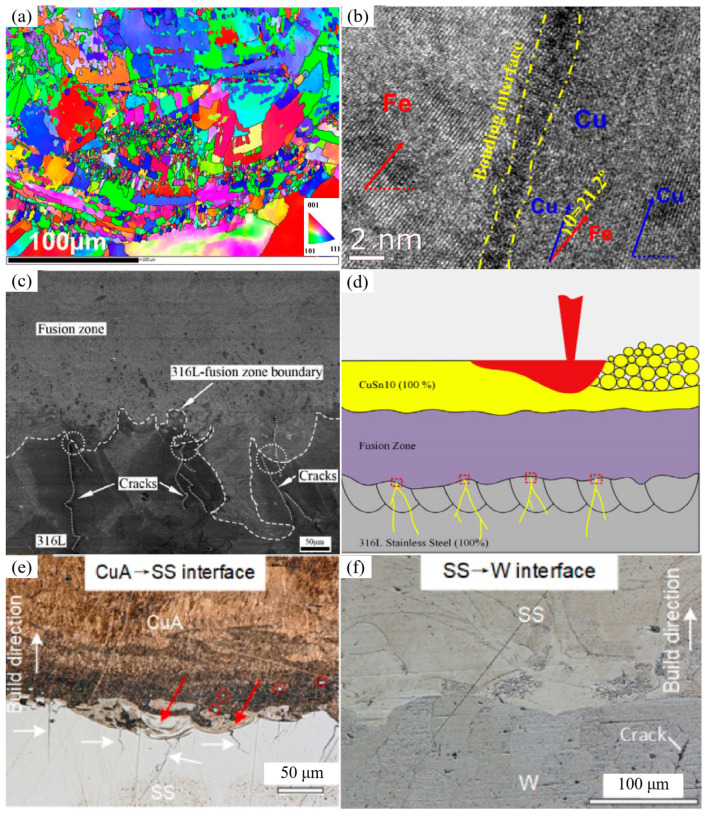
(**a**) Inverse pole figure of interfacial microstructure; (**b**) High-resolution transmission electron microscope (HRTEM) image of the bonding interface; (**c**) Crack distribution in the fusion zone; (**d**) Schematic diagram of the dendritic cracks; (**e**) The interface of CuA/316L, red arrows representing elemental mixing, white arrows representing cracks; (**f**) Interface of 316L/W [94,103,104].

**Figure 7 materials-16-02757-f007:**
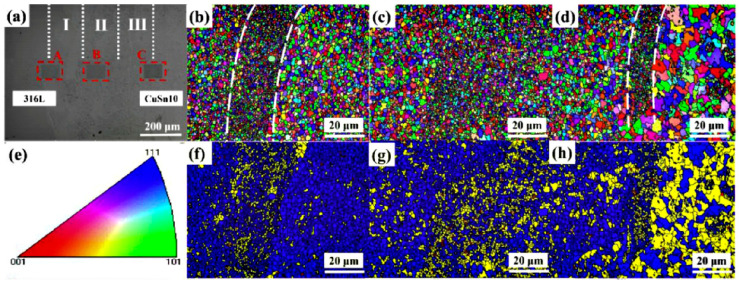
LPBF-processed 316L/CuSn10 specimen, (**a**) SEM image, I, II, III are three main interfacial transion regions, A is the region from 316L to the transition zone, B is the area in the transition zone, C is the area from the transition zone to the CuSn10 alloy; (**b**–**d**) EBSD orientation maps from A, B and C regions marked by the red dotted frame; (**e**) IPF; (**f**–**h**) Recrystallisation distribution [105].

**Figure 8 materials-16-02757-f008:**
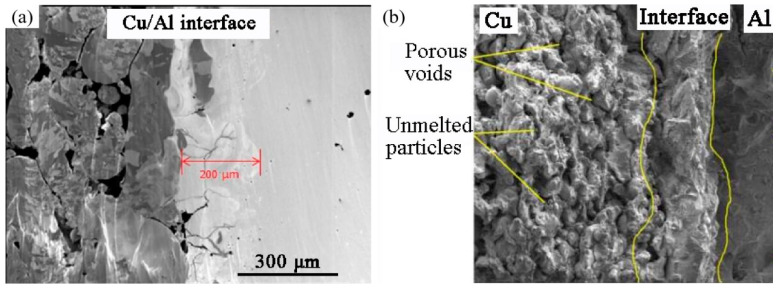
(**a**) Focused ion beam image of LPBF-processed Cu/Al interface; (**b**) Fracture morphology [92].

**Figure 9 materials-16-02757-f009:**
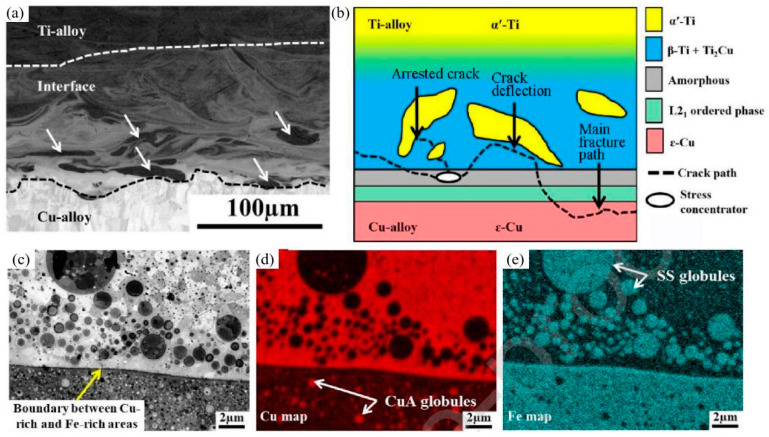
(**a**) Ti–Cu interface, arrows indicate regions of *α*’-Ti phase within a matrix of *β*-Ti + Ti_2_Cu; (**b**) Crack propagation paths under tensile load; (**c**) Fe/Cu interface; (**d**,**e**) Cu and Fe elemental map [93].

**Figure 10 materials-16-02757-f010:**
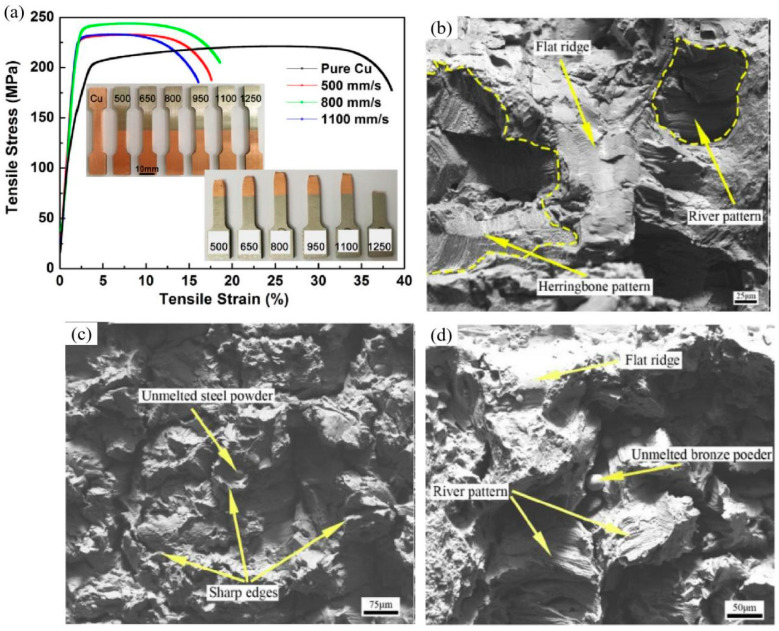
(**a**) Tensile stress–strain curves produced by different processing parameters; (**b**) The fusion-zone fracture morphology of horizontal combination; (**c**,**d**) CuSn10 side and 316L side fracture morphologies [103,104].

**Figure 11 materials-16-02757-f011:**
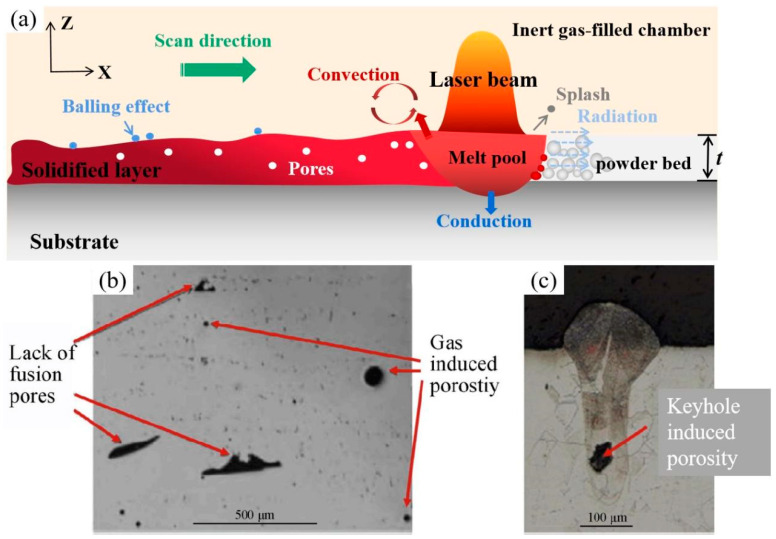
(**a**) Schematic overview of the interaction zone between the laser and powder bed; (**b**) Lack of fusion and gas-induced porosity [140]; (**c**) Keyhole-induced porosity in the LPBF process [139].

**Figure 12 materials-16-02757-f012:**
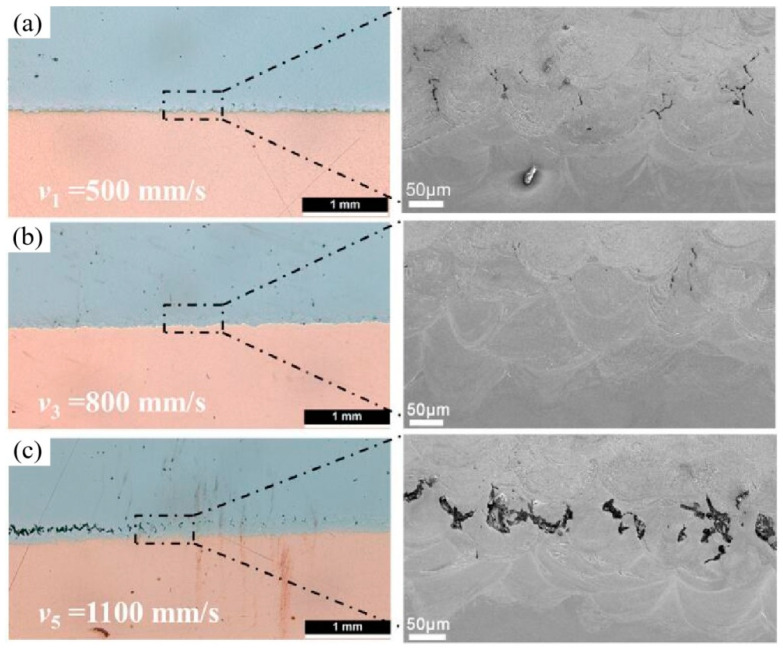
Interfacial defects in the vertical cross sections of LPBF-processed copper–steel specimens observed by optical micrographs and SEM, (**a**–**c**) indicate the interfacial bonding morphology when scan speed (*v*) is 500 mm/s, 800 mm/s and 1100 mm/s respectively [103].

**Figure 13 materials-16-02757-f013:**
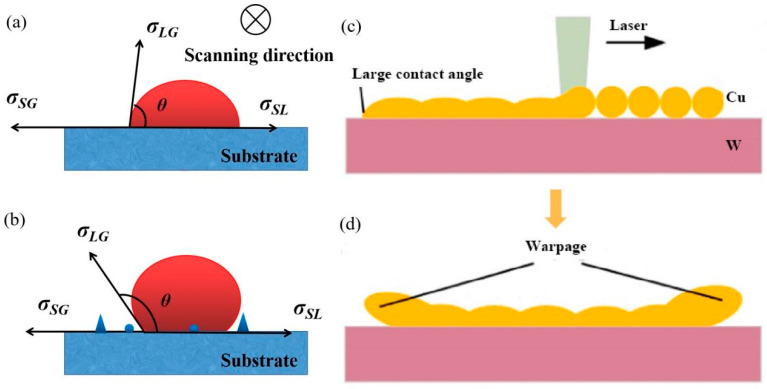
The contact angle of a droplet: (**a**) On a smooth surface; (**b**) On a coarse surface; (**c**) Cu directly bonding to W; (**d**) The warpage induced by poor wettability [94].

**Figure 14 materials-16-02757-f014:**
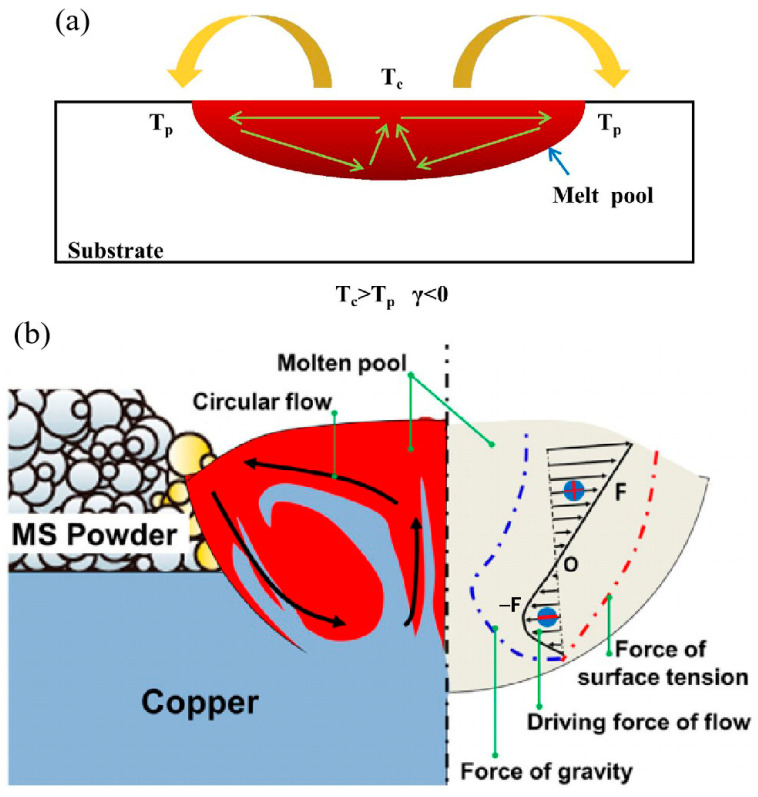
Schematic of (**a**) Marangoni convection induced by surface tension; (**b**) Corresponding formation mechanism in melt pool at the interface of maraging steel/Cu [103].

**Figure 15 materials-16-02757-f015:**
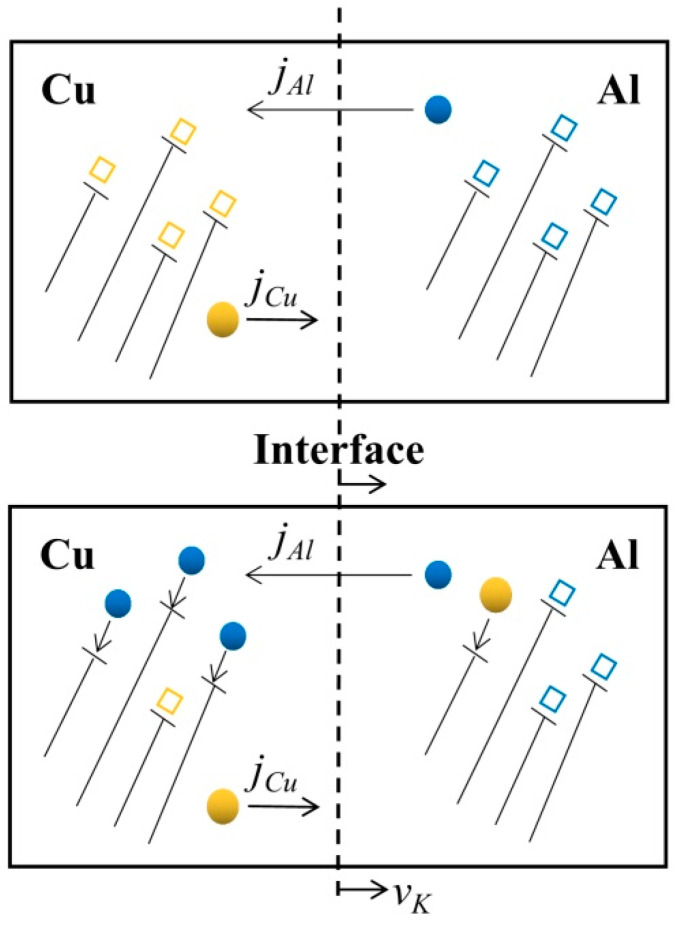
Kirkendall effect dominates element diffusion [170].

**Figure 16 materials-16-02757-f016:**
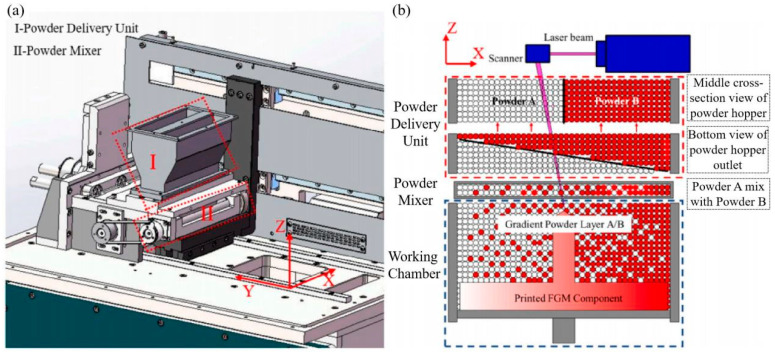
(**a**) Schematic of the self-developed compositionally-graded alloys LPBF system; (**b**) 2D section view of powder-delivery unit and mixer [175].

**Figure 17 materials-16-02757-f017:**
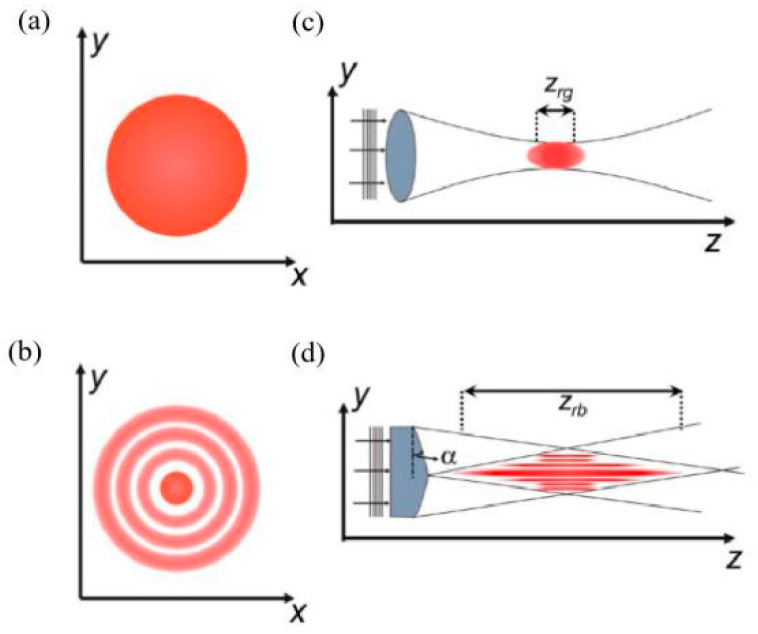
Schematic of (**a**) Gaussian; (**b**) zero-order Bessel beam profiles in the x-y plane; (**c**,**d**) represent the z-axis propagation and focusing properties of Gaussian and Bessel beams [180].

**Table 1 materials-16-02757-t001:** Summary of mechanical properties of LPBF-processed dissimilar metal materials.

Combination	Materials (Powders)	Mechanical Properties				Refs
		Tensile Strength (MPa)	Yield Strength (MPa)	Elongation (%)	Interface Hardness	
Al–Al	Al12Si, Al3.5Cu1.5Mg1Si	369 ± 15	267 ± 10	2.6 ± 0.1		[90]
Al–Cu	AlSi10Mg, C18400	176 ± 31				[92]
Cu–Fe	Cu10Sn, 316L SS (vertically combined)	423.3 ± 30.2		4.6 ± 0.9	212.8 ± 7.6 HV	[104]
	Cu10Sn 316L SS (horizontally combined)	451.9 ± 8.0		10.5 ± 1.7		
	Cu10Sn, 316L SS (vertically combined)	418.9 ± 13.5		6.2 ± 1.4	301.9 ± 17.9 HV (I region)	[105]
					223.1 ± 10.9 HV (II region)	
					203.8 ± 8.3 HV (III region)	
	Cu10Sn, 18Ni300	144.4 ± 41.59		1.59 ± 0.524	130.8 ± 5.93 Gpa	[117]
		128.7 ± 7.55 Remelting twice		2.41 ± 0.529	123.4 ± 2.69 Gpa	
Fe–Ni	316SS IN718	596 ± 10		28.1 ± 2		[118]
Ti–Fe	Ti6Al4V, K220Cu, 316L SS	523 ± 1 (V650)				[93]
		496 ± 23 (V500)				
		473 ± 26 (V350)				
Ti–Ti	Ti5Al2.5Sn, Ti6Al4V	1034		about 4.5%		[119]
W–Cu	Pure W, CuA				172.67 HV	[94]
Fe–Fe	Maraging steel, H13	664		about 23%	about 630 HV_50_	[89]

**Table 2 materials-16-02757-t002:** Further potential applications of LPBF-processed dissimilar metal materials.

Combination	Application	Physical Properties
Cu/Ni	Rocket combustor Aerospace industries	Thermal conductivity, high-temperature resistance
Cu/W	Heat sink for high-power integrated circuits High-performance electrodes Plasma-facing components of nuclear fusion reactors	Plasma-radiation resistance, thermal conductivity, electrical conductivity
Cu/Al	Heat sink Rail transport	Low density, thermal conductivity, electrical conductivity
Cu/Fe	Heat exchangers Conformal cooling channel in the mould	Electrical conductivity, wear resistance, thermal conductivity, high stiffness
Ti/Fe	Nuclear container Aerospace industries	Corrosion resistance, mechanical properties
Ti/Ta	Bone implants	Biocompatibility, corrosion resistance, nontoxic, mechanical properties
Ti/Al	Submarine shell	Corrosion resistance, lightweight, mechanical properties

## Data Availability

The authors can provide raw or processed data upon request.

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
