# Peer review of "Laser Powder Bed Fusion of Dissimilar Metal Materials: A Review"

_materials, 2023, doi:10.3390/ma16072757_

Round 1
Reviewer 1 Report
The authors provide a review of the current state of laser powder bed fusion (LPBF) techniques. This review is a good overview of the current state of the field. I do not understand the use of the word "promising" at the end of the abstract. However, as this area is rapidly evolving, various points of view and reviews are warranted.
There are many figures which has small text. These figures should be made larger to ensure readability. Many of these figures are reproductions from the literature and may require permissions.
Pg. 2 - First Paragraph: No reference is made to studies using different build atmospheres.
Pg. 5 - First Paragraph: "... cooling rate ... provides sufficient thermal activation ...". This is a confusing. Please clarify.
Pg. 5 - First Paragraph: "The temperature controlled by E ..." There appears to be a typo. What is "E"?
Pg. 5 - End of First Paragraph: Add some spaces to the equations to make them easier to read. Where is the reference for this discussion?
Pg. 5 - Last 2 Paragraphs: Is this discussion all from reference [102]. Is figure 4 from [102]? Is there a formatting issues with equation 1 (missing comma and an indent which can be removed)?
Pg. 6 - First Paragraph: It is not obvious that the joint is "flawless". Please rephrase.
Pg. 7 - First Sentence: "steel copper". Typo?
Pg. 7: Define "CuA"
Pg. 8 - Figure 6 Caption: Define "HRTEM"
Pg. 8 - Figure 7 Caption: Define "EBSD"
Pg. 9 - 3rd Line: Type in references; extra space.
Pg. 9: Typo - "L21" should be "L2_{1}"
Pg. 9 - Last Sentence: "SS /CuA" should be "SS/CuA"; remove the extra space.
Pg. 10 - Second Paragraph: Typo "Table 1.", remove the period after the table number.
Pg. 11 - Table 1: Clean up formatting. Consider landscape vs portrait.
Pg. 12 - Last Paragraph: "as equation (1) [127]" should be "as equation (1) of [127]" or it has been left out(?).
Pg. 13 - First Paragraph: "copper–stainless steel" should be "Cu-SS" following previous usages.
Pg. 15 - Equation 2: This equation is out of place.
Pg. 15 - First Paragraph: No description of "recoater".
Pg. 15 - Equation 3: Improper formatting.
Pg. 15 - Second Paragraph: "[52], [145,157]" should be " [52, 145,157]"
Pg. 15 - Third Paragraph: " penetrate sufficiently the previously deposited" should be " penetrate sufficiently to the previously deposited"
Pg. 16 - Equation 4: Improper formatting. Also, m, k, and T should be italicized in the text matching the equation, and the definition of T is missing from the text.
Pg. 16 - Equation 5: Fix formatting.
Pg. 17 - First Paragraph: "Nguyen" should be "Nguyen et al.".
Pg. 18 - Second Line: "Part 2" to what is this referring?
Pg. 18 - Last Paragraph: "E" should be italicized. Are there references missing? Defining the variables listed would help the non-expert reader.
Pg. 19 - First Paragraph: Machine learning techniques are helpful, but they do not always yield the underlying physics and chemistry. High throughput experiments decrease the per-sample cost, but may not decrease the budget of a particular investigation. A bit more detail here would be good.
Pg. 19 - Second Paragraph: "316 ss" should be "316 SS".
Pg. 19 - Second Paragraph: This paragraph discusses both laser wavelength and mode (Gaussian vs Bessel). Are there references missing in regards to wavelength and the absorptivity of the powders?
Pg. 19 - Section 6.3: Why isn't Ti/Ta and Ti/Al listed in the table with other applications?
Pg. 20 - Conclusion #2: "E" should be written out or should refer back to the section in which it is defined.
Author Response
Reviewer 1:
The authors provide a review of the current state of laser powder bed fusion (LPBF) techniques. This review is a good overview of the current state of the field. I do not understand the use of the word "promising" at the end of the abstract. However, as this area is rapidly evolving, various points of view and reviews are warranted.
Thanks a lot. We revised “promising” to “significant”.
There are many figures which has small text. These figures should be made larger to ensure readability. Many of these figures are reproductions from the literature and may require permissions.
Thanks for your suggestions. We have made figures and the text in figures be larger, such as Fig. 9, Fig.10, Fig.11 and Fig.16. Also, we provided the permissions of figures, which are reproduction from the literature, as seen in word document “Permissions”. In addition, the Ref[170] is my own work.
Pg. 2 - First Paragraph: No reference is made to studies using different build atmospheres.
Thanks a lot. We have provided the Ref[8] “Effect of metal vaporization behavior on keyhole-mode surface morphology of selective laser melted composites using different protective atmospheres. Apll. Surf. Sci. 2015, 355, 310-319”.
Pg. 5 - First Paragraph: "... cooling rate ... provides sufficient thermal activation ...". This is a confusing. Please clarify.
Thank you very much. In fact, we referred to the literature [87], which mentioned that “,high energy inputs from either laser or electron beam heat sources can provide enough thermal activation energy to form unique microstructures and phases, which are unachievable through equilibrium cooling” . We delete the sentence “In non-equilibrium cooling”.
Pg. 5 - First Paragraph: "The temperature controlled by E ..." There appears to be a typo. What is "E"?
Thanks a lot. The “E” is “laser energy density”. We have revised in the manuscript.
Pg. 5 - End of First Paragraph: Add some spaces to the equations to make them easier to read. Where is the reference for this discussion?
Thanks very much. We have added some spaces to the equations. In addition, this discussion derived from basic solidification theory. We referred a chinese textbook and this theory can be known by the researchers.
Pg. 5 - Last 2 Paragraphs: Is this discussion all from reference [102]. Is figure 4 from [102]? Is there a formatting issues with equation 1 (missing comma and an indent which can be removed)?
Thanks very much. This discussion also derived from basic solidification theory. We have revised equation (1). The Fig.4 is not from [102], but is self-drawn.
Pg. 6 - First Paragraph: It is not obvious that the joint is "flawless". Please rephrase.
Thanks a lot. We have revised the “flawless” to “well-connected”.
Pg. 7 - First Sentence: "steel copper". Typo?
Thanks very much. We have revised to “steel/copper”.
Pg. 7: Define "CuA"
Thanks a lot. “CuA” means “Cu alloy”. We have defined it in the manuscript.
Pg. 8 - Figure 6 Caption: Define "HRTEM"
Thank you for your advice. We have added “High resolution transmission electron microscope” in Fig.6 to define “HRTEM”
Pg. 8 - Figure 7 Caption: Define "EBSD"
Thank you for your advice. We have added “Electron backscattered diffraction” in Fig. 5. So we still use “EBSD”in Fig.7.
Pg. 9 - 3rd Line: Type in references; extra space.
Thanks a lot. We deleted the extra space.
Pg. 9: Typo - "L21" should be "L2_{1}"
Thanks a lot. We changed “1” to subscript.
Pg. 9 - Last Sentence: "SS /CuA" should be "SS/CuA"; remove the extra space.
Thank you. We removed the extra space.
Pg. 10 - Second Paragraph: Typo "Table 1.", remove the period after the table number.
Thanks a lot. We have removed the period.
Pg. 11 - Table 1: Clean up formatting. Consider landscape vs portrait.
We reformatted the format of Table 1.
Pg. 12 - Last Paragraph: "as equation (1) [127]" should be "as equation (1) of [127]" or it has been left out(?).
Thank you for pointing out the problem. It should be “as equation (2) of [127].
Pg. 13 - First Paragraph: "copper–stainless steel" should be "Cu-SS" following previous usages.
Thanks a lot. We have revised.
Pg. 15 - Equation 2: This equation is out of place.
Thanks a lot. We have put it in proper place.
Pg. 15 - First Paragraph: No description of "recoater".
Thank you. The “recoater” is the structure to spread powders in LPBF process. The recoater is equal to scraper. We revised it to “scraper” in the manuscript.
Pg. 15 - Equation 3: Improper formatting.
Thank you. In fact, we used the formula editor to format equation 3.
Pg. 15 - Second Paragraph: "[52], [145,157]" should be " [52, 145,157]"
Thank you very much. We have revised the similar problems in whole texts.
Pg. 15 - Third Paragraph: " penetrate sufficiently the previously deposited" should be " penetrate sufficiently to the previously deposited"
Thanks. We have added.
Pg. 16 - Equation 4: Improper formatting. Also, m, k, and T should be italicized in the text matching the equation, and the definition of T is missing from the text.
Thanks a lot. The m, k and T have been italicized. T is the temperature of the melt, which can be known from “The energy input affects the temperature gradients between the centre and the edge of the melt pool, as shown in Fig. 14(a). As the energy input increases, the viscosity of the melt decreases, which promotes flowability. The relationship between viscosity (μ) and temperature (T) can be defined as follows”.
Pg. 16 - Equation 5: Fix formatting.
Thanks. We have used the formula editor to format equation 5.
Pg. 17 - First Paragraph: "Nguyen" should be "Nguyen et al.".
Thank you. We have added.
Pg. 18 - Second Line: "Part 2" to what is this referring?
Thank you. To avoid ambiguity, we deleted “, as stated in Part 2”.
Pg. 18 - Last Paragraph: "E" should be italicized. Are there references missing? Defining the variables listed would help the non-expert reader.
Thank you very much. We have revised “E” to be italicized and defined it in the manuscript.
Pg. 19 - First Paragraph: Machine learning techniques are helpful, but they do not always yield the underlying physics and chemistry. High throughput experiments decrease the per-sample cost, but may not decrease the budget of a particular investigation. A bit more detail here would be good.
Thanks very much for your suggestion. We added some sentence.
Pg. 19 - Second Paragraph: "316 ss" should be "316 SS".
Thanks very much. We have revised.
Pg. 19 - Second Paragraph: This paragraph discusses both laser wavelength and mode (Gaussian vs Bessel). Are there references missing in regards to wavelength and the absorptivity of the powders?
Thank you for your suggestion. The Part 6 is the “ Future trends and perspectives”. Different kinds of powders have different absorptivity to laser in same or different wavelength. So we proposed that “If the wavelength range can be matched to the laser absorption of different metallic materials, it is expected that dissimilar components with defect-free, good interfacial joints and excellent service properties can be produced based on Bessel beams”.
Pg. 19 - Section 6.3: Why isn't Ti/Ta and Ti/Al listed in the table with other applications?
Thanks very much. We have listed in Table 2.
Pg. 20 - Conclusion #2: "E" should be written out or should refer back to the section in which it is defined.
Thank you very much. We have revised to “laser energy density” and defined it.
Reviewer 2 Report
The authors have presented a review of research outcomes in the field of LPBF of dissimilar materials. Especially the influence of processing factors at the dissimilar materials as well as effect of microstructure of fusion zone on mechanical properties of the build part is reviewed. The area of research is interesting and the review is concluded well. Here are few comments which might help in improving the manuscript.
1. Figure 1 does not clearly illustrates what the authors want to show and why. The figure may be improved.
2. In first sentence at p-4, reference of studies may be added.
3. In "temperature controlled by E in front of....." at p-5, E is not defined. Define all such alphabets used in the paper.
4. Further, in 1st para of section 3, angle between growth direction of grains and melt pool are mentioned. IT is recommended to draw a diagram to illustrate such directions and other parameters.
5. It is recommended to mention if equation (1) is impirical. If not give a reference of source of equation.
6. Alignment of figures may be kept uniform and corrected wherever needed.
7. Equations are not referred in the text which they should be.
8. The future trends and perspectives are not written clearly. They seem to be a literature review again but not future trends. Authors may review section 6 accordingly.
Author Response
The authors have presented a review of research outcomes in the field of LPBF of dissimilar materials. Especially the influence of processing factors at the dissimilar materials as well as effect of microstructure of fusion zone on mechanical properties of the build part is reviewed. The area of research is interesting and the review is concluded well. Here are few comments which might help in improving the manuscript.
- Figure 1 does not clearly illustrates what the authors want to show and why. The figure may be improved.
Thanks very much for your suggestion. About Fig. 1, we referred to the literature [87] “Additive manufacturing of metals: Microstructure evolution and multistage control”, which is published in Journal of Materials Science & Technology in 2022, to state some possible combinations of dissimilar metal materials. We improved the resolution.
- In first sentence at p-4, reference of studies may be added.
Thank you very much. We added Ref[101] in first sentence.
- In "temperature controlled by E in front of....." at p-5, E is not defined. Define all such alphabets used in the paper.
Thank you for pointing out the deficiency. We have defined “E” as “laser energy density”.
- Further, in 1st para of section 3, angle between growth direction of grains and melt pool are mentioned. IT is recommended to draw a diagram to illustrate such directions and other parameters.
Thanks very much. We have added diagram in Fig.4(a) to illustrate.
- It is recommended to mention if equation (1) is impirical. If not give a reference of source of equation.
Thanks very much. The equation (1) derived from basic solidification theory. We referred a chinese textbook and this theory can be known by the researchers
- Alignment of figures may be kept uniform and corrected wherever needed.
Thanks very much for your suggestion. We have revised the figures.
- Equations are not referred in the text which they should be.
Thanks very much. We have put equation (2) to proper site.
- The future trends and perspectives are not written clearly. They seem to be a literature review again but not future trends. Authors may review section 6 accordingly.
Thanks very much. We combined some literatures as basis to make perspectives. From some existing possible solutions and key problems, the future research direction of LPBF-processed dissimilar metal materials could be proposed.
Reviewer 3 Report
Dear Authors, I have read the paper carefully and found it very informative and well written. I recommend minor revision, with just one suggestion and one clarification:
Section 4, 2nd sentence appears unfinished, please rewrite.
Figure 13: So, small wetting angle theta results in large contact angle on the deposited layer?
Thank you
Author Response
Dear Authors, I have read the paper carefully and found it very informative and well written. I recommend minor revision, with just one suggestion and one clarification:
Section 4, 2nd sentence appears unfinished, please rewrite.
Thanks very much for pointing out the problem. We added “,which is beneficial to nucleation”.
Figure 13: So, small wetting angle theta results in large contact angle on the deposited layer?
Thanks very much. The contact angle is used to express wettability between deposited layer and next layer. As known from Young's relation, small wetting angle theta would lead to a reduction in solid-liquid interface energy and an increase in wettability. High wettability results in strong bonding area of solid A and solid B.
Thank you
Round 2
Reviewer 2 Report
All suggestions are well addressed by authors in the revised manuscript.